# Benefit of Hyaluronic Acid to Treat Facial Aging in Completely Edentulous Patients

**DOI:** 10.3390/jcm11195874

**Published:** 2022-10-04

**Authors:** Selene Aubry, Pierre-Yves Collart-Dutilleul, Matthieu Renaud, Dominique Batifol, Sylvie Montal, Laurence Pourreyron, Delphine Carayon

**Affiliations:** 1Centre de Soins, d’Enseignement et de Recherche Dentaires CSERD, Centre Hospitalier Universitaire de Montpellier, 34193 Montpellier, France; 2Faculty of Dentistry, University of Montpellier, 34193 Montpellier, France; 3Laboratory Bioengineering Nanosciences LBN, University of Montpellier, 34193 Montpellier, France; 4Maxillo-Facial Surgery Department, Centre Hospitalier Universitaire de Montpellier, 34295 Montpellier, France

**Keywords:** complete denture, edentulism, facial aging, hyaluronic acid, lip support

## Abstract

Hyaluronic acid (HA) is widely used in aesthetic medicine for its moisturizing and anti-aging action. This molecule, which is naturally present in the body, has an interesting response to aging, accentuated in totally edentulous patients. While its aesthetic benefits for facial rejuvenation are well-documented, there is a lack of description and investigation on its therapeutic usefulness for edentulous patients. The management of completely edentulous patients is a daily reality in dental practice and requires specific attention. The aesthetic and functional challenge is considerable. The displacement of the bone base, which is often marked, and lack of soft tissue support are sometimes difficult to correct with prosthetic reconstruction. This review aims to present the physiological processes appearing in completely edentulous patients and prosthetic solutions available to recreate oral functions and counteract facial aging. As prosthetic rehabilitations are not fully satisfying for counterbalancing the impression of excessive facial aging, we investigated the applications of HA injection in the perioral area, in order to improve edentulism treatment, and discussed the advantages and disadvantages, compared to other dermal fillers and rejuvenation therapies. Considering the specific situations of edentulous patients, dermal HA injections help to correct uncompensated bone losses and mucous volume losses and appear to be a therapeutically beneficial for treating completely edentulous patients, without the requirement to full rejuvenation therapy.

## 1. Introduction

Facial aging is a natural and unavoidable phenomenon, with more visible effects for completely edentulous patients. The transition to edentulousness represents physical, psychological, and social handicaps and should be considered with empathy, with the objective to restore patient’s smile.

Edentulism is easily assimilated to aging, but it has also been proved to be associated with depression and poor self-rated health [1]. Tooth loss reflects the endpoint of dental disease. In 2010, 158 million people (2.3% of the global population) were completely edentulous. Even though the prevalence of severe tooth loss reduced between 1990 and 2010, declining from 4.4% to 2.3%, the burden of edentulism is likely to grow as populations age and live longer [2,3,4]. Nowadays, aesthetic expectations have increase among people of all ages [5]. Clinical situations and patients’ expectations and complaints will guide smile restoration towards fixed or removable prosthetic rehabilitation. Understanding the specificity of aging in total edentulous patients is essential for overcoming facial changes and restoring an aesthetic and functional smile. Loss of volume in the lower third of the face is inherent in aging, due to elastic and collagenous fibers degeneration, fat tissue reduction, and skin dehydration. This volume loss is aggravated by edentulism, as teeth play an important role in lip and cheek support, and tooth loss is sometimes not completely compensated by prostheses [6,7,8]. The limitations of complete dentures in restoring tissue loss, as well as in fully supporting the lips and cheeks, will lead to the appearance of premature ageing. Facial muscles may lose some of their tone through ageing, but dystonia may also occur because of functional limitations, as the underlying artificial dentures are only sitting on the mucosa and not attached to the facial skeleton [9]. The labial contour of the denture determines the amount of support provided for the lip, and inadequate lip and cheek support induces a poor facial appearance.

Therefore, it is of prime importance to understand the specific physiopathology of facial aging occurring with edentulism. To counteract the excessive impression of facial aging in edentulous patients, dental and orofacial treatments can target both prosthodontics and lip dynamism and volume. Different methods can be used for non-surgical rejuvenation and beautification of the lower third of the face, such as microfocused ultrasound, radiofrequency, botulinum toxin, and injectable fillers [10]. The therapeutic choice will be directed towards a specific product or technique, according to the desired effect, properties of the product, duration of action, and practitioner’s experience [11]. For tissue volume and dynamism, different injectable fillers can be used, such as calcium hydroxyapatite (CaHA), permanent silicon, collagen, or hyaluronic acid (HA). Hyaluronic acid is the most popular biodegradable dermal filler, widely used in aesthetic medicine for facial rejuvenation and soft tissue volume augmentation. It is a natural polysaccharide found in human dermis and epidermis. When injected in dermis, its effect lasts from 6 to 18 months, depending on the source, extent of cross-linking, and concentration [12]. In addition to volume augmentation, HA plays a major role in connective tissues hydration, improves tissue vascularization, and activity stimulates dermal fibroblasts [13].

The aim of this review is to discuss the potential therapeutic role of hyaluronic acid for restoring edentulous patients’ smiles. Considering the challenges to overcome with completely edentulous patients’ rehabilitation, starting from clinical cases, then literature survey, we investigate the specificity of aging in completely edentulous patients and therapeutic possibilities for treating edentulism. We discuss the interest of hyaluronic acid injections for improving and perfecting prosthetic restoration.

## 2. Completely Edentulous Patients Physiological Processes

### 2.1. Bone Resorption

Bone structures are the support for skin tissues. Over time, they undergo changes related to general factors, such as aging, osteoporosis, or possible deficiencies in calcium absorption, as well as local factors, such as dental extraction, occlusal trauma, lack of adaptation of prostheses, and uncompensated tooth loss over a prolonged period, causing repercussions on the supported tissues. These multiple parameters vary accordingly for every patient because they are dependent on individual genetic variations. The main changes are a decrease in the anterior height of the facial mass, correlated with tooth loss and alveolar bone resorption [6]. With aging, bone structures undergo morphological and structural changes, such as involution and atrophy. The loss of teeth results in the absence of bone stimulation through chewing and, therefore, a decrease in bone density and loss of width and height of the alveolar bone [14]. Regeneration and augmentation of the alveolar bone could be necessary for aesthetic and functional prosthetic restoration [15].

In the first year after tooth loss, the alveolar bone is reduced by about 25% of its width, and resorption in the anterior region is generally four times greater in the mandible than in the maxilla. The maxillary incisor–canine region benefits from the presence of several skin muscles that limit anterior resorption. In the maxillary posterior sectors, alveolar bone resorption concerns mainly the vestibular sides of the ridges and occurs in a centripetal manner. Conversely, in the posterior mandibular sectors, resorption is centrifugal, accentuating the anteroposterior displacement of the maxillae [14,16]. In addition, the height of the jawbone is reduced, due to the expansion of the maxillary sinus, which continues over time, involving a decrease in the vertical dimension and crushing of the profile [17,18,19]. The chin appears to be fully projected forward, which is responsible for the typical profile of total edentulism (Figure 1) [10].

### 2.2. Soft Tissues Aging

Lips can be described as two parts—one white, cutaneous, and peripheral, and the other one red, at the inner part, called vermilion. This red part is divided into a dry visible part and wet internal part. Lips are bounded by nasal ala, nasolabial folds, and labiomental fold. In a completely edentulous patient, the facial muscles inserted on structural bones are no longer supported during their contractions because of tooth loss.

Among these, the orbicularis oris muscle encircles the mouth, inside the lips. A lack of support induces mouth orbicular muscle atrophy, then loss of volume and lips eversion. As a consequence, the visible part of the vermilion shrinks, and the lips invaginate inside the oral cavity, accentuating fine lines and wrinkles on and around the lips. Lowering muscles at the mouth angle (depressor anguli oris) and depressant muscles of the lower lip (depressor labii inferioris) accentuates the labiomandibular folds. The nasolabial angle increases, and the labiomental crease is more pronounced (Figure 2) [8,20,21,22].

## 3. Prosthetic Solutions in Response to Aging in Completely Edentulous Patients

Bone structures resorption undeniably leads to an increase in prosthetic height. In the horizontal direction, the antero-posterior shift of bone structures complicates the restoration of aesthetics and function [23,24].

Depending on prosthetic rehabilitation, a prospective set-up will allow to visualize and foreshadow the future prosthesis, in order to validate aesthetics and function. During prosthesis realization, the steps of occlusion model adjusting and teeth positioning are essential, as they will directly influence the labial projection and lip positions.

### 3.1. Importance of Occlusal-Vertical Dimension (OVD)

In completely edentulous patients, the determination of adequate maxillo–mandibular relationship in three dimensions is the first and most important step [25].

For the vertical dimension, an underestimation of the height of the lower part of the face will accentuate the already marked nasolabial and labio-mental folds and generate salivary stasis that might induce candidiasis, such as angular cheilitis. Previous prostheses can guide the practitioner in OVD determination. However, anterior and posterior landmarks have often disappeared [26]. Then, it is a matter of defining an agreement fitting for both physiology and aesthetics. A reduction of OVD will induce an increase of skin folds and inadequate lip support.

### 3.2. Importance of Prosthetic Teeth Positioning

The support of the upper lip is controlled by maxillary incisors coronal two-thirds [27]. The patient’s type and profile guide the practitioner in the choice of incisors shape (triangular, square, or oval). Their size, position on the dental arch, and bulging condition the lips support. The fitting of prosthetic teeth is meticulously checked from front and profile [28]. Front view allows us to adjust the symmetry and position of the maxillary central incisors, axial inclination, and ratio between teeth. The profile view guides parallelism of the curves, contour of the incisal edge, and passive labial support [29,30].

Prosthetic teeth are positioned according to functional, aesthetic, and phonetic requirements. Thin or prominent lips are more affected by incisal edge position than thick or vertical lips. An overly buccal incisor profile is an obstacle to lip closure. In the profile view, the harmony of the lower third of the face correlates with S-shaped curves, following the sub-nasal point, white-red lip junction, and chin tip.

### 3.3. Artificial Gingiva

During completely edentulous patient rehabilitation (either fixed on implants or removable), prosthetic teeth play a central supporting role for the lips, but are not sufficient to compensate the recoil of the orbicular muscle (orbicularis oris) above the cranial part of the prosthesis.

Peri-oral soft tissues support can be corrected by means of the polished exterior surfaces of the prostheses, which have both functional and aesthetic roles [31]. Muscles perpendicular to the occlusion plane of the prosthesis have a destabilizing role. On the contrary, muscle fibers parallel to this plane act positively on the stability of the prosthesis. When modeling the artificial gingiva, a concave profile is sought next to orbicular muscle, where the fibers are parallel to the occlusion plane [32,33].

Artificial gingiva also provides support for the lip: its shape and volume are an integral part of the aesthetic success of the smile. In a large-scale anterior rehabilitation, the resin thickness of a fixed prosthesis will compensate bone loss and lack of support from the overlying soft tissues [8]. In addition, at the level of the maxillary canine bulges, which undergo strong bone resorption after extraction, the resin sculpted in the axis of the prosthetic teeth will compensate alveolar processes collapse and lift the nasolabial fold and labial commissure (Figure 3).

Even when carefully respecting the theoretical and clinical rules of occlusal vertical dimension and teeth positioning, the prosthetic rehabilitations cannot fully counteract the impression of excessive facial aging, with lack of lip support, limited soft tissue volume, and marked folds. Therefore, there is a need for therapeutic intervention on the perioral soft tissue, in order to improve edentulism treatment.

## 4. Hyaluronic Acid: Therapeutically for Dental Reconstruction and Lip Support

In modern societies, influenced by advertising and social networks, aesthetic demands and patients’ expectations are constantly increasing. The quest for youth is growing considerably and pushes dental surgeons to become architects of smiles.

In western culture, dental prosthesis integration should be as natural as possible. However, a prosthesis itself cannot always respond to patient’s expectations and complaints. Smile conception and design will aim not only at functionality, but also at aesthetics, in harmony with the face.

### 4.1. Hyaluronic Acid (HA)

Hyaluronic acid (HA) is a molecule naturally produced in human body by various cells (fibroblast, synovial cells, muscle cells, and endothelial cells). HA is a natural polysaccharide and component of the human dermis and epidermis. It harbors many properties, such as hydration and suppleness of the skin, healing, intra-articular lubrication, and antioxidants, making it essential for the human body for skin suppleness and joints functioning. Synthetic hyaluronic acid injections are used in many medical fields: derivatives of HA are the biodegradable fillers most widely used in Europe and the USA [34].

Injection of HA fillers in the peri-oral area (lip, nasolabial and labiomandibular folds, and labiomental crease) adds a tool to the therapeutic arsenal for the completely edentulous patient, when prostheses are not sufficient to compensate for too advanced bone loss. These peri-oral injections should be considered part of oral rehabilitation treatment.

Products are conditioned as gel in syringes, with different viscosities, depending on the hyaluronic acid concentration (Figure 4).

Viscosity will be chosen according to the area to be injected: more viscous for nasolabial folds and chin and more fluid for lips and small ridules. Several techniques are available, depending on the area to be corrected: retrotracing injections using an atraumatic cannula or multipuncture injections with a needle. It is imperative to inject with the prostheses positioned in the mouth to avoid over-correction and distortion of the final result.

### 4.2. Particularity of Completely Edentulous Patients

Each anatomic region differs slightly, in terms of injection technique. The goal in every region is to avoid danger zones that could lead to skin necrosis or visual loss [35]. Marked nasolabial folds can be filled with HA to restore volume and unfold the tissues (Figure 5). Drooping labial commissures, giving “a sad look”, can be relieved by injection below the end of the commissure (Figure 6).

Lips are often very thin in completely edentulous people, despite wearing prostheses. In such situation, HA fillers are of prime interest for volumizing effects that can last for 6 to 18 months, depending on the source, the extent of the cross-linking and concentration of injected product [12,36]. It is particularly indicated for restoring the natural curves of lips and redefining the vermilion contour (Figure 7A,B).

In addition, the presence of asymmetry and eversion of lips will make it even more complex to rehabilitate the smile in complete toothless situations, in which soft tissues are the only reference points [37]. Wrinkle filling and soft tissue augmentation can correct such defects and harmonize the smile line with the lip curvature (Figure 8).

Following sometimes traumatic avulsions, canine bumps are frequently affected by increased resorption affecting the soft tissues by a profile hollowed out below the wings of the nose. Prosthetic extrados are not always sufficient for restoring this volume. Volumizing injections in these areas are likely to correct marked defects.

When the bone shift is not fully compensated by the wearing of prostheses, hyaluronic acid injections prove to be a tool that should not be neglected for the restoration of volume to the various tissues and compensation of the shift in the jawbones.

It is undoubtedly recommended to perform the injections with the prostheses in the mouth, in order to be able to anticipate and optimize the desired result, while avoiding unsightly over-corrections, which may subsequently cause discomfort during the placement of the prostheses (Figure 9).

## 5. Discussion

Starting from clinical challenges, we aimed to highlight the interest of hyaluronic acid in the total edentulous patient in general practice, through clinical cases and literature survey. There are many studies detailing the use and safety of soft-tissue fillers, but none of them investigated the interest of HA as a therapeutically of prosthetic restorations in the fully edentulous patient [35].

As rightly stated, “no one filler is the correct choice for all application” [37]. However, due to its volumizing and hydrating properties, low risk of complications, resorbability of the molecule, and simplicity of use, HA appears to be an appropriate filler for reducing the nasiolabial fold, redefine lips contour, and improve live support in edentulous patient, with an obvious beneficial effect in the aging face [35].

There are many available fillers that can be used for dermal injection, thus providing volumetric support. Animal collagen and synthetic fillers (hyaluronic acid, calcium hydroxyapatite, polylactic acid, and polymethylmethacrylate) have volumetric and biological effects, such as effects on fibroblast production and angiogenesis. Fat grafting and autologous platelet-derived preparations are also interesting techniques for facial rejuvenation because they are autologous and have regenerative effects [38].

Calcium hydroxyapatite fillers (composed by CaHA microspheres surrounded by resorbable aqueous gel carrier) and polylactic acid are interesting for correcting deep wrinkles and folds, with an immediate result after the injection, but the lack of immediately reversible still represents a major limitation to complement full denture rehabilitation, when the effects of misplaced or excess of HA can be reversed within few days. In the same vein, the main problems of polymethylmethacrylate injections are the long-term, questionable side effects and permanent duration [11].

Regenerative procedures could also be considered to complement full denture rehabilitation, in order to have a more proactive prevention and maintenance approach. Indeed, regenerative approaches in facial rejuvenation are the logical steps in aging treatment. These procedures can include stem cell use, autologous platelet-derived preparations (i.e., PRF), or fat grafts [39].

Autologous platelet-derived preparations for dermal injection have been shown to lead to significant rejuvenation of the face skin [40]. The rejuvenation properties of fat grafts and fat-derived stem cells have also been described, as well as the intradermal injections of fat, combined with platelet-rich fibrin (PRF) for patients undergoing facial rejuvenation treatments [41]. The regenerative effects of adipose tissue and cell enriched fat grafts on facial aesthetics were clear at the histologic and cellular levels, but these regenerative effects were not clinically apparent when comparing cell enriched fat grafts to fat grafts alone. Both fat-PRF and HA injections were shown to improve facial skin status without serious complications [41]. Fat grafts and PRF or HA injections are recognized to be safe, highly effective, and long-lasting methods for skin rejuvenation. Fat-grafts and PRF show higher biological effects, but require invasive sampling (blood or fat tissue), while HA is a ready-to-use device.

In daily dental practice, we observe an increase in the number of total edentulous patients already carrying oral rehabilitation with classical full denture or complex implant-supported rehabilitations. These restorations can be functionally satisfactory, while no longer fulfilling their role of supporting soft tissues, due to inevitable aging. In such circumstances, there is a need for volume and tissue support to complement the dental prosthetic rehabilitation. Among all the available fillers and regenerative procedures, HA injections appear to be an optimal compromise between efficacy, simplicity, and safety. HA can then be a satisfactory solution to improve aesthetics without repairing prostheses and entering into full rejuvenation treatments.

## 6. Conclusions

The growing aesthetic demand of patients, of all ages, requires being attentive to their requests and responding in an appropriate manner. The democratization of youthfulness in the media and society imposes on practitioners, as well as on patients, a permanent cosmetic approach. HA allows us to improve patients’ smiles. For specific situations of edentulous patients, dermal HA injections combine the cosmetic aspect with the therapeutic aspect. These minimally invasive injections help to correct, in an ephemeral way, the uncompensated bone losses, mucous volume losses, and asymmetries. In addition to the dental rehabilitation of complete edentulism, and without the requirement to full rejuvenation therapy, HA injections are a beneficial therapeutically to treat completely edentulous patients.

## Figures and Tables

**Figure 1 jcm-11-05874-f001:**
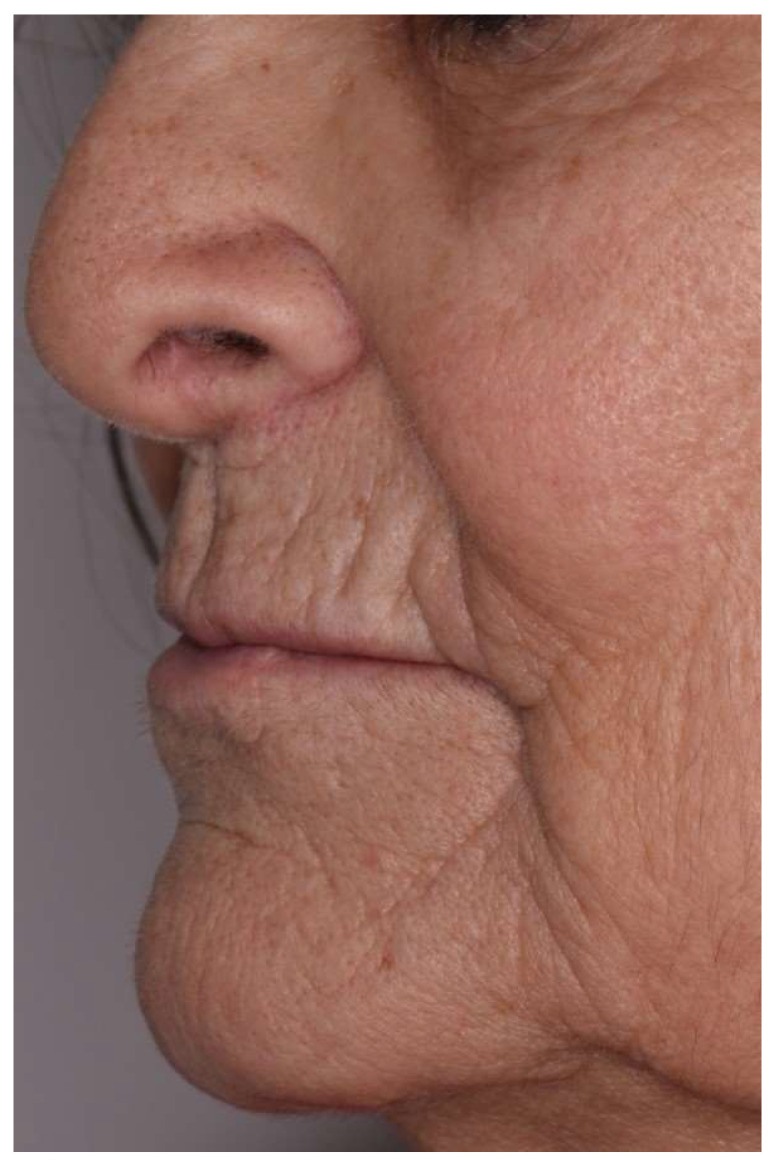
Typical profile of the total edentulous patient, without prosthetic restoration: the chin is projected forward, and the lips are thin and crushed.

**Figure 2 jcm-11-05874-f002:**
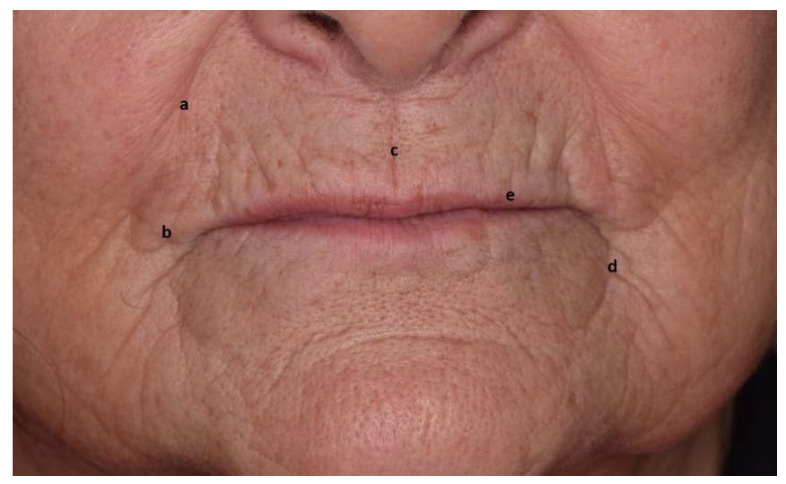
“Scope of action” of the dental practitioner. The lower third of the face: (a) nasolabial fold, (b) labial commissure, (c) philtrum, (d) bitterness fold, and (e) vermilion zone.

**Figure 3 jcm-11-05874-f003:**
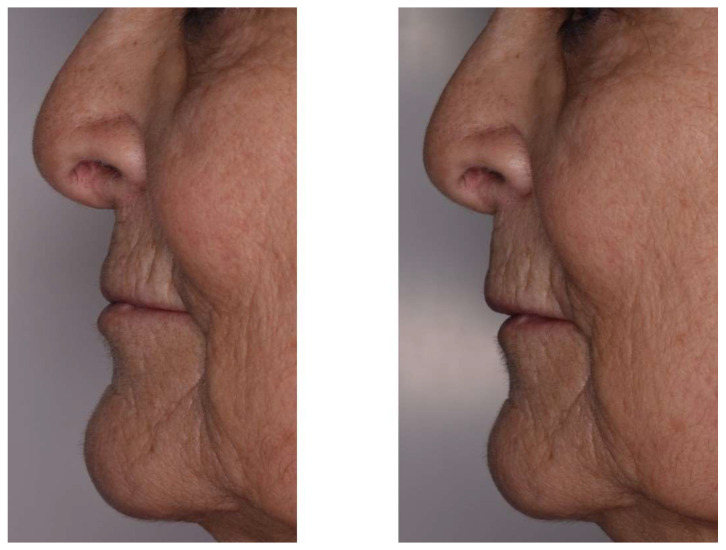
Patient profile without and with removable prosthesis with artificial gingiva.

**Figure 4 jcm-11-05874-f004:**
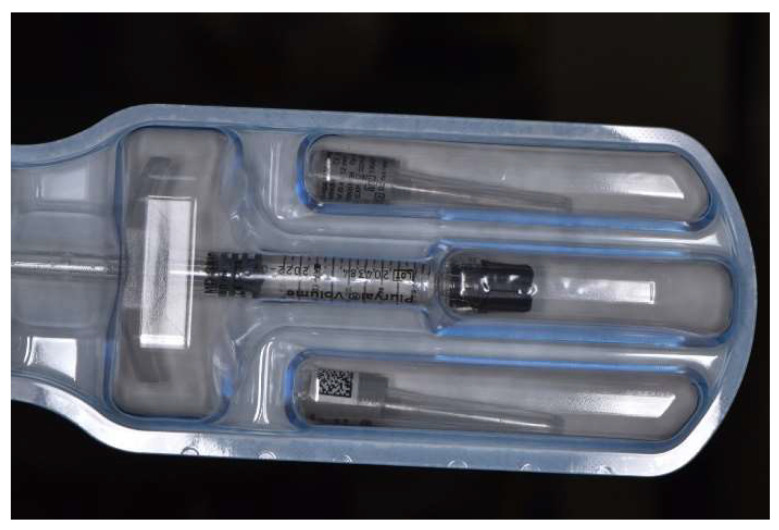
Packaging mode of the pre-filled syringe of hyaluronic acid.

**Figure 5 jcm-11-05874-f005:**
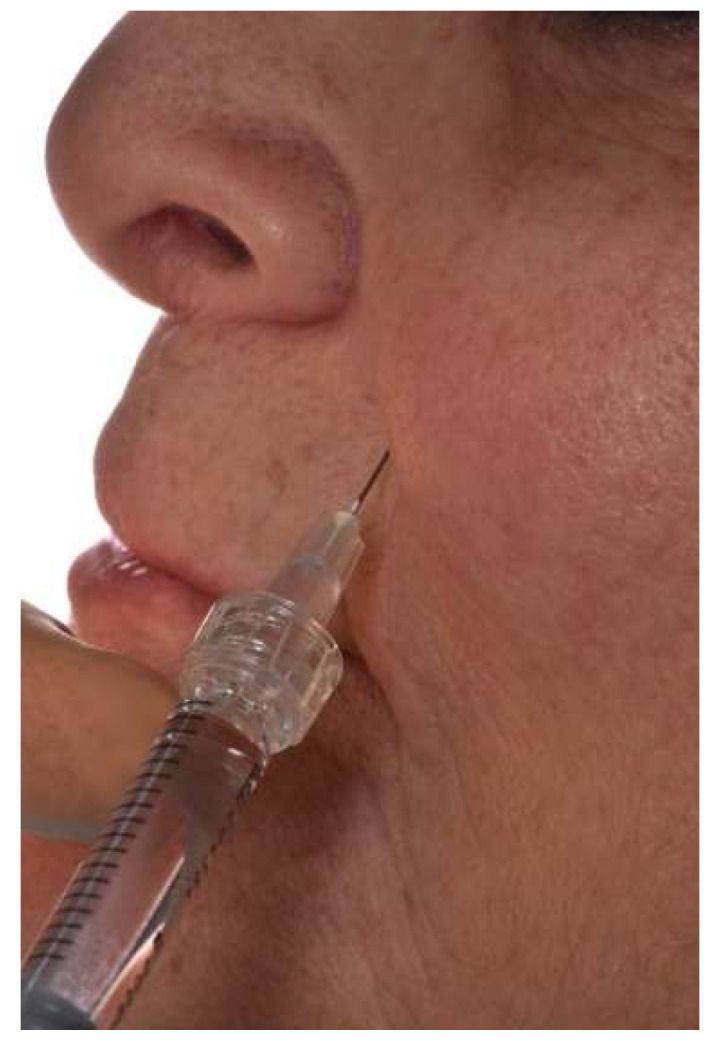
Injection of hyaluronic acid in front of the nasolabial folds in a patient rehabilitated by complete bimaxillary prostheses.

**Figure 6 jcm-11-05874-f006:**
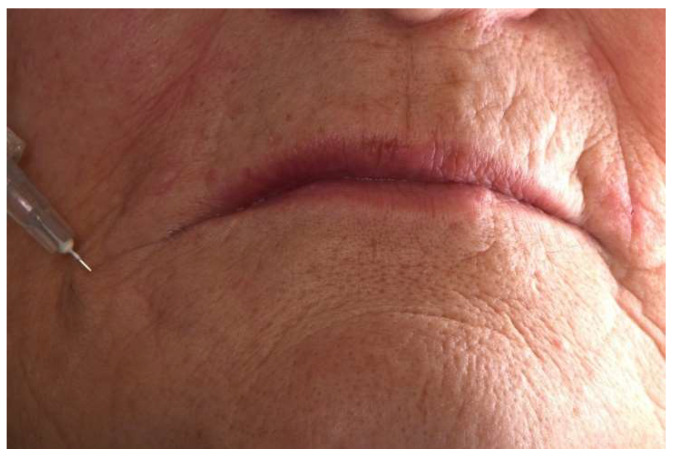
Injection of hyaluronic acid in front of the labial commissures.

**Figure 7 jcm-11-05874-f007:**
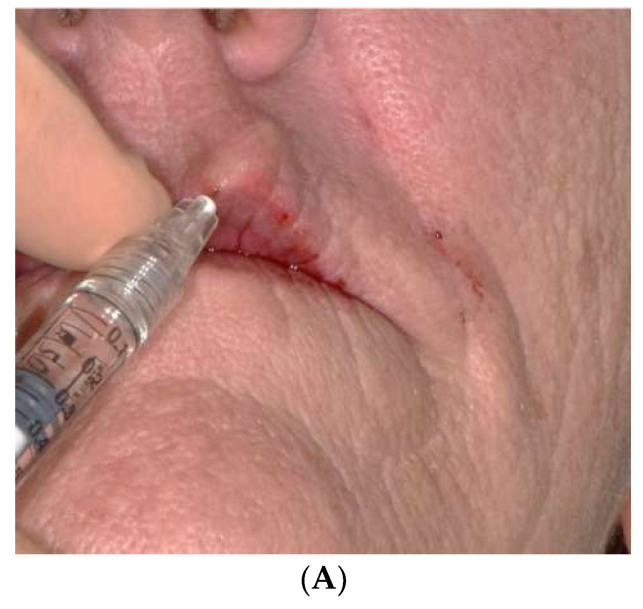
(**A**). Injection of hyaluronic acid in front of the upper lip. (**B**). Pre- and post-injection front view of a bimaxillary full denture patient seeking to reshape and redefine the lip contour.

**Figure 8 jcm-11-05874-f008:**
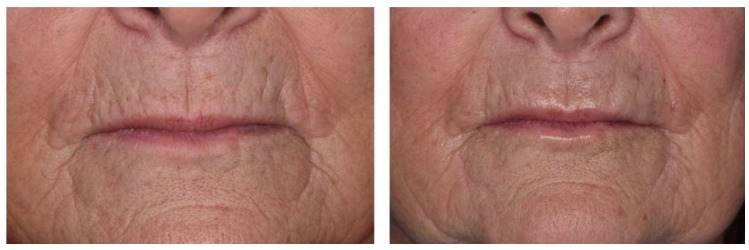
Front view before and after injection of a patient with asymmetry and eversion of the upper lip. The solution considered with the patient consisted of injecting hyaluronic acid, in unequal quantities, into the upper right and left lips.

**Figure 9 jcm-11-05874-f009:**
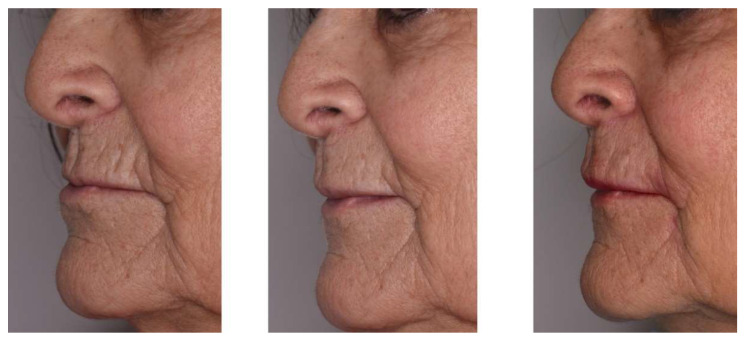
Profile view without prosthesis, with removable complete prosthesis before and immediately after hyaluronic acid injection.

## Data Availability

Not applicable.

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
