# Peer review of "Benefit of Hyaluronic Acid to Treat Facial Aging in Completely Edentulous Patients"

_jcm, 2022, doi:10.3390/jcm11195874_

Round 1

Reviewer 1 Report

This paper aims to underline the interest of hyaluronic acid to remedy the effects of aging in edentulous patients.

The abstract was written as a prologue and must be rewritten as an abstract.

The introduction is not focused on HA, despite providing some related facts this part fails to support the aim.

The rest of the paper is poorly written, many statements do not include cites and (apparently) are based on authors' experiences, no relevant discussion about literature was included, and conclusions are not supported by the results.

Author Response

Reviewer 1: Answers to Comments and Suggestions

This paper aims to underline the interest of hyaluronic acid to remedy the effects of aging in edentulous patients.

We thank the reviewer for all their comments and suggestions and we have tried to emphasize the interest of the article based on his comments.

The abstract was written as a prologue and must be rewritten as an abstract.

We have rewritten the abstract to emphasize the objective of the manuscript, which is in fact, in addition to the literature review, essentially based on our clinical cases. Many publications are made on the interest of injections in aesthetic medicine but none on the fact that these injections are therapeutic allies in the prosthetic treatment of totally edentulous patients, which is the originality of this paper.

The introduction is not focused on HA, despite providing some related facts this part fails to support the aim.

We agree with this comment. However, as the article is mainly intended to highlight the specificity of edentulous patients, we have added in the introduction the existence of other filling methods and have specified that we have used hyaluronic acid in our clinical practice. Its action is mainly described in part 4. Still, we have completed the ‘Introduction’ section to support the objective of the manuscript.

The rest of the paper is poorly written, many statements do not include cites and (apparently) are based on authors' experiences, no relevant discussion about literature was included, and conclusions are not supported by the results.

We have taken this comment into account: we have revised the whole manuscript and added a discussion paragraph. We focused more on literature survey and put our clinical experience and interpretation at a second plan (additional references throughout the text). All modifications, corrections and added part are written in red in the revised manuscript.

Reviewer 2 Report

The review was aimed to discuss benefits of hyaluronic acid to reduce facial aging effects. However, the authors didn't provide any suitable comparison and discussion of hyaluronic acid  with other existing antiaging treatment: SPRS- and MST/SVF- therapy, injections of microspheres, collagen etc.

Discussing the importance of bone and gingival restoration the authors completely ignored the role of regenerative medicine and the most recent publications on this topic. When writing about the bone resorption the authors did not compare autogenous and synthetic bone grafts. The authors should extend the discussion comparing the advantages and weaknesses of different treatment options and conclude the review with suitable recommendations regarding the application of HA. 

The list of references should be extended at least with novel and most important publications, highlighting role of regenerative approaches:

Kulakov, A., Kogan, E., Brailovskaya, T., Vedyaeva, A., Zharkov, N., Krasilnikova, O., ... & Klabukov, I. (2021). Mesenchymal stromal cells enhance vascularization and epithelialization within 7 days after gingival augmentation with collagen matrices in rabbits. Dentistry journal9(9), 101.

Smirani, R., Rémy, M., Devillard, R., & Naveau, A. (2022). Use of Human Gingival Fibroblasts for Pre-Vascularization Strategies in Oral Tissue Engineering. Tissue Engineering and Regenerative Medicine19(3), 525-535..

Yazdanian, M., Arefi, A. H., Alam, M., Abbasi, K., Tebyaniyan, H., Tahmasebi, E., ... & Rahbar, M. (2021). Decellularized and biological scaffolds in dental and craniofacial tissue engineering: A comprehensive overview. journal of materials research and technology15, 1217-1251.

Recommended improvements will help the article to meet the requirements of scientific journal.

Author Response

Reviewer 2: Answers to comments and suggestions

The review was aimed to discuss benefits of hyaluronic acid to reduce facial aging effects. However, the authors didn't provide any suitable comparison and discussion of hyaluronic acid with other existing antiaging treatment: SPRS- and MST/SVF- therapy, injections of microspheres, collagen etc.

We thank the reviewer for this comment. According to the literature review, we have added in the introduction and then in the discussion other filling methods. We have rewritten the abstract to emphasize the objective of the manuscript, which is in fact, in addition to the literature review, essentially based on our clinical cases. Many publications are made on the interest of injections in aesthetic medicine but none on the fact that these injections are therapeutic allies in the prosthetic treatment of totally edentulous patients, which is the originality of this paper. The article is mainly intended to highlight the specificity of edentulous patients, but we have added, in the introduction, the existence of other filling methods and have specified that we have used hyaluronic acid in our clinical practice.

Discussing the importance of bone and gingival restoration the authors completely ignored the role of regenerative medicine and the most recent publications on this topic. When writing about the bone resorption the authors did not compare autogenous and synthetic bone grafts.

We thank the reviewer for this comment but we feel that this is outside the scope of the article.  However, following this note, we have added a sentence about the need for bone grafts in certain clinical cases : ….”Regeneration and augmentation of the alveolar bone could be necessary for esthetic and functional prosthetic restoration. (Yazdanian et al, 2021)”…

The authors should extend the discussion comparing the advantages and weaknesses of different treatment options and conclude the review with suitable recommendations regarding the application of HA. 

We followed the reviewer's advice and added a discussion paragraph. We have focused on literature survey (additional references throughout the text). All modifications, corrections and added part are written in red in the revised manuscript.

The list of references should be extended at least with novel and most important publications, highlighting role of regenerative approaches:

We thank the reviewer for these references and have added the following references in the text (added references are written in red in the manuscript):

  1. Braz, A.; Eduardo, C.C. de P. Reshaping the Lower Face Using Injectable Fillers. Indian J Plast Surg 2020, 53, 207–218, doi:10.1055/s-0040-1716185.
  2. Dayan, S.H.; Ellis, D.A.F.; Moran, M.L. Facial Fillers. Facial Plastic Surgery Clinics of North America 2012, 20, 245–264, doi:10.1016/j.fsc.2012.05.004.
  3. La Gatta, A.; Salzillo, R.; Catalano, C.; D’Agostino, A.; Pirozzi, A.V.A.; De Rosa, M.; Schiraldi, C. Hyaluronan-Based Hydrogels as Dermal Fillers: The Biophysical Properties That Translate into a “Volumetric” Effect. PLoS ONE 2019, 14, e0218287, doi:10.1371/journal.pone.0218287.
  4. Yazdanian, M.; Arefi, A.H.; Alam, M.; Abbasi, K.; Tebyaniyan, H.; Tahmasebi, E.; Ranjbar, R.; Seifalian, A.; Rahbar, M. Decellularized and Biological Scaffolds in Dental and Craniofacial Tissue Engineering: A Comprehensive Overview. Journal of Materials Research and Technology 2021, 15, 1217–1251, doi:10.1016/j.jmrt.2021.08.083.
  5. Rohrich, R.J.; Bartlett, E.L.; Dayan, E. Practical Approach and Safety of Hyaluronic Acid Fillers. Plastic and Reconstructive Surgery - Global Open 2019, 7, e2172, doi:10.1097/GOX.0000000000002172.
  6. Bass, L.S. Injectable Filler Techniques for Facial Rejuvenation, Volumization, and Augmentation. Facial Plastic Surgery Clinics of North America 2015, 23, 479–488, doi:10.1016/j.fsc.2015.07.004.
  7. Crowley, J.S.; Kream, E.; Fabi, S.; Cohen, S.R. Facial Rejuvenation With Fat Grafting and Fillers. Aesthetic Surgery Journal 2021, 41, S31–S38, doi:10.1093/asj/sjab014.
  8. Cohen, S.R.; Hewett, S.; Ross, L.; Delaunay, F.; Goodacre, A.; Ramos, C.; Leong, T.; Saad, A. Regenerative Cells For Facial Surgery: Biofilling and Biocontouring. Aesthetic Surgery Journal 2017, 37, S16–S32, doi:10.1093/asj/sjx078.
  9. Hassan, H.; Quinlan, D.J.; Ghanem, A. Injectable Platelet‐rich Fibrin for Facial Rejuvenation: A Prospective, Single‐center Study. J Cosmet Dermatol 2020, 19, 3213–3221, doi:10.1111/jocd.13692.
  10. Liang, Z.-J.; Lu, X.; Li, D.-Q.; Liang, Y.-D.; Zhu, D.-D.; Wu, F.-X.; Yi, X.-L.; He, N.; Huang, Y.-Q.; Tang, C.; et al. Precise Intradermal Injection of Nanofat-Derived Stromal Cells Combined with Platelet-Rich Fibrin Improves the Efficacy of Facial Skin Rejuvenation. Cell Physiol Biochem 2018, 47, 316–329, doi:10.1159/000489809.

Round 2

Reviewer 1 Report

This new version have included many improvements. However, abstract is still a prologue. 

A second point is about the purpose of section 3, which is not clearly connected to HA.

Author Response

Reviewer 1: Answers to Comments and Suggestions_2nd Revision

This new version have included many improvements. However, abstract is still a prologue.

We thank you again for re-reading our manuscript. We have modified and rewritten the abstract, to avoid this aspect of being a prologue. (modifications in red in the revised manuscript)

A second point is about the purpose of section 3, which is not clearly connected to HA.

Indeed, the section 3 is not directly connected to Hyaluronic Acid. This section describes the guidelines for edentulous rehabilitation, with their inherent limitations, especially lack of lip support. Efficient denture will restore the ability to chew but limited lip support and impression of excessive facial aging affects appearance, self-esteem and self-confidence. Therefore HA injections (or, eventually, other dermal fillers) have an important interest for patients’treatments. We have added a short paragraph, at the end of section 3, to explain this link between dental rehabilitation and HA injections:

“Even when carefully respecting the theoretical and clinical rules of Occlusal-Vertical Dimension and teeth positioning, the prosthetic rehabilitations cannot fully counteract the impression of excessive facial aging, with lack of lip support, limited soft tissue volume, and marked folds. Therefore, there is a need for therapeutic intervention on the perioral soft tissue, in order to improve edentulism treatment.”
